# SHINE: Shielding Backdoors in Deep Reinforcement Learning

## Abstract

Recent studies have discovered a deep reinforcement learning (DRL) policy is vulnerable to backdoor attacks. Existing defenses against backdoor attacks either do not consider RL's unique mechanism or make unrealistic assumptions, resulting in limited defense efficacy, practicability, and generalizability. We propose SHINE, a backdoor shielding method specific for DRL. SHINE leverages policy explanation techniques to identify the backdoor triggers and designs a policy retraining algorithm to eliminate the impact of the triggers on backdoored agents. We theoretically prove that SHINE guarantees to improve a backdoored agent's performance in a poisoned environment while ensuring its performance difference in the clean environment before and after shielding is bounded. We further conduct extensive experiments that evaluate SHINE against three mainstream DRL backdoor attacks in various benchmark RL environments. Our results show that SHINE significantly outperforms existing defenses in mitigating these backdoor attacks.

## 1 Introduction

Deep reinforcement learning has demonstrated remarkable performance in various sequential decision-making problems, ranging from beating professorial human players in Go (Silver et al., 2016) and real-time strategy games (DeepMind, 2017; OpenAI, 2017) to controlling robots to accomplish sophisticated tasks (Levine et al., 2016; Tai et al., 2017). Along with its great success comes a new security concern of the supply chain management of DRL agents – backdoor threats. Specifically, recent research (Kiourti et al., 2019; Wang et al., 2021a) demonstrates that an attacker could train a backdoored agent and outsource it to a user. After the user deploys the agent in the corresponding environment, the attacker places the backdoor trigger in the environment, forcing the agent to take non-optimal actions and thus reduce its total reward. Defending backdoor attacks in DRL is inherently challenging in that the trigger pattern is typically imperceptible, and the backdoored agent performs normally at clean states. Based on different trigger patterns, existing backdoor attacks against DRL can be categorized as perturbation-based attacks and adversarial agent attacks (detailed in Section 2).

Existing defenses against backdoor attacks consider two setups – 1) training-phase defense that trains a robust model from a poisoned dataset (e.g., Tran et al. (2018); Du et al. (2019); Weber et al. (2020); Zhang et al. (2022); Wu et al. (2022); Zhang et al. (2022)) and 2) testing-phase defense that shields a pretrained model/policy (e.g., Wang et al. (2019); Gao et al. (2019); Chou et al. (2020); Ma et al. (2022); Wang et al. (2022); Guo et al. (2022); Bharti et al. (2022)). We consider the second setup to shield a pretrained (backdoored) agent from being affected by the backdoor trigger presented in the environment. Most existing testing-phase defenses are designed for supervised classifiers (e.g., Wang et al. (2019); Guo et al. (2020); Liu et al. (2019); Gao et al. (2019); Chou et al. (2020); Ma et al. (2022); Wang et al. (2022)). Due to the significant difference between DRL agents and classifiers in the mechanism (i.e., sequential decision-making vs. individual class prediction) and model output (i.e., action at each time step vs. predicted class), these techniques either cannot be applied or demonstrate very limited efficacy in defending backdoors in DRL agents. Only a few defenses are designed specifically for DRL agents (Bharti et al., 2022; Guo et al., 2022). As we will discuss in Section 2, Bharti et al. (2022) design their defense only for the perturbation-based attack, while the defense proposed in Guo et al. (2022) is only applicable to adversarial agent attacks.

In this work, we propose SHINE, a novel method to shield a pretrained DRL agent against both the perturbation-based attacks and the adversarial agent attacks. Technically, we first collect a set of

trajectories of the agent running in the environment and design a two-stage explanation method to identify the backdoor trigger presented in these trajectories. Our explanation method first pinpoints the states where the trigger is most likely to be presented. Then, it identifies a common subset of features in the trigger-presenting states that are most critical to the agent's action at these states and deems them as the backdoor trigger. With the identified trigger, we then design a policy retraining algorithm to shield the agent from being affected by the backdoor. By carefully designing the retraining objective function, we theoretically guarantee that the retrained agent performs better in a poisoned environment while maintaining its performance in the pristine clean environment.

We evaluate SHINE against three most prevalent backdoor attacks (perturbation-based attacks in single-agent RL, adversarial agent attacks in two-player competitive RL, and perturbation-based attacks in multi-agent cooperative RL) in seven benchmark RL environments. Our results demonstrate that SHINE is effective against all three attacks in terms of faithfully identifying the trigger and improving a backdoored agent's performance in a poisoned environment. Additionally, we demonstrate that SHINE outperforms two representative defenses designed for supervised classifiers (Wang et al., 2019; 2022) and a state-of-the-art DRL defense (Bharti et al., 2022). Second, we show that SHINE does not jeopardize (or even improve) a *clean* agent's performance. This is an essential property in that users could apply SHINE to arbitrary agents without worrying about the negative impact on the clean ones. Finally, we verify that SHINE retains its effectiveness against attacks with different variations (e.g., simple and complex triggers) and possible adaptive attacks. To the best of our knowledge, this is the first backdoor defense against both perturbation-based and adversarial agent attacks in both single- and multi-agent RL that does not require accessing a clean environment.

## 2  RELATED WORK

**Backdoor Attacks.** Based on different trigger patterns and injection methods, existing attacks can be categorized as: 1) perturbation-based attack that uses a perturbation patch as the trigger (Kiourti et al., 2019; Wang et al., 2021b; Chen et al., 2022b) and 2) adversarial agent attack that uses an adversarial agent's certain actions as the trigger (Wang et al., 2021a). Based on the target environment, existing attacks can be further categorized as attacks against single-agent RL (Yang et al., 2019; Kiourti et al., 2019; Wang et al., 2021b), attacks against two-agent competitive RL (Wang et al., 2021a; Chen et al., 2022a), attacks against multi-agent cooperative RL (Chen et al., 2022b;a).

**Backdoor Defenses.** As mentioned in Section 1, we focus on the testing-phase defense, and most existing works in this category target supervised classifiers (e.g., Wang et al. (2019); Guo et al. (2020); Liu et al. (2019); Gao et al. (2019); Chou et al. (2020); Ma et al. (2022); Wang et al. (2022)). As we will show in Section 4, due to the fundamental differences between RL and supervised learning, these techniques demonstrate limited efficacy in DRL backdoor defense. Only a few research works have focused specifically on mitigating backdoors in DRL agents. Unfortunately, these methods have limited practicability and generalizability. Specifically, the defense proposed in Bharti et al. (2022) is designed only for perturbation-based attacks, and it is only applicable to environments with a discrete action space. The method in Guo et al. (2022) is designed particularly for adversarial agent attacks. Different from existing methods, our defense is applicable to both perturbation-based attacks and adversarial agent attacks. In addition, we demonstrate its efficacy in all three types of target environments mentioned above. Note that recent works also extend backdoor attacks and defenses to weak-supervised learning (Saha et al., 2022; Carlini & Terzis, 2021; Yan et al., 2021) and federated learning (Bagdasaryan et al., 2020; Wang et al., 2020; Xie et al., 2019), which are beyond our scope.

Our technique is inspired by the DRL explanation methods that identify the agent's critical state and actions (Amir & Amir, 2018; Huang et al., 2018; Jacq et al., 2022; Guo et al., 2021b). Although they help pinpoint the time steps when the trigger is likely to be presented, These methods can neither identify the trigger nor shield the target agent from the backdoor attack.

## 3  KEY TECHNIQUE

### 3.1  OVERVIEW

**Threat Model.** We consider perturbation-based and adversarial agent attacks, where perturbation-based attacks add the trigger to the agent's perceived state without changing the actual state and

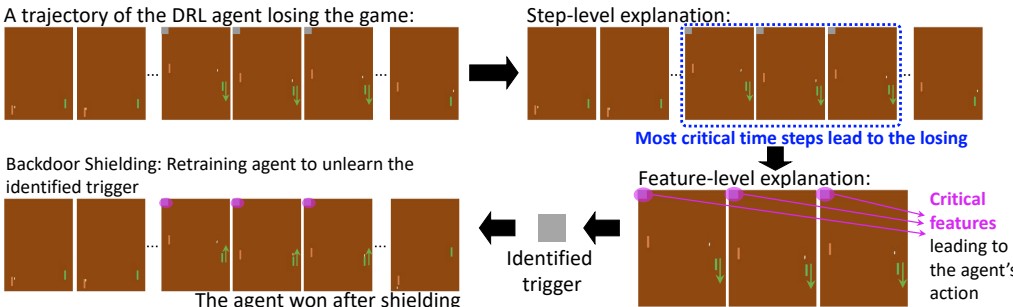

Figure 1: Overview of SHINE. The green paddle is the DRL agent and the arrow indicates its action.

its transition. By contrast, adversarial agent attacks use an agent's action sequences as the trigger, manipulating the actual state and the state transition. We are given a pretrained agent operating in a potentially poisoned environment. The agent (denoted as shielding agent) has a policy $\pi$ that potentially contains a backdoor. We assume the state and action space can be discrete or continuous.

Rather than assuming the defender has a clean environment that cannot be accessed or poisoned by the attacker, we assume the attacker and defender access the same environment that the agent actually operates (denoted as operating environment). After the agent is deployed, the attacker can place the trigger at any desired states to spitefully distract the backdoored agent and reduce its reward. As the defender, we do not assume any knowledge of the trigger and thus cannot decide whether it is present.

We believe this is a practical setup because it simulates the actual RL operation scenarios where the environment is a natural scene or a simulator created by a third party, and the attacker and defender have the same privilege to access the environment. Note that different from supervised learning, where it is relatively easy to hold out a clean validation set for the defender, constructing a clean environment could be extremely difficult in DRL. Take the self-driving car as an example. To build a clean environment, the defender must construct a simulator of the actual road scenarios and traffic conditions. The amount of effort required is often beyond the capacity of a policy user, necessitating the involvement of a specialized third party. Once released, the simulator becomes accessible to authorized users, both benign and malicious.

**Technical Overview.** Our *goal is to ensure the shielding DRL agent can normally perform, regardless of whether the trigger is presented in the environment*. Take Fig. 1 as an example. The attacker can present the trigger (small patch at the top left corner) at any desired time steps, and we want to ensure the agent can still obtain a decent winning rate in the poisoned/operating environment and the original clean environment. At a high level, SHINE has a *trigger restoration* step to identify the backdoor trigger, followed by a *backdoor shielding* step to prevent our shielding agent from being affected by the backdoor. More specifically, for the trigger restoration, we draw insights from the DRL explanation methods (Amir & Amir, 2018; Huang et al., 2018; Jacq et al., 2022; Guo et al., 2021b) and identify the trigger from the previous trajectories of the shielding agent. The idea is as follows (demonstrated in Fig. 1). Given the backdoor attack's goal is to fail a victim agent by distracting its actions with the backdoor trigger. Here, failing means the agent receives a very low reward or loses the game. By analyzing and explaining why a backdoored agent failed in its previous game rounds (trajectories), we should be able to identify the key reason, i.e., the trigger presented in the environment. In particular, given a failing trajectory of our shielding agent, we first utilize a step-level explanation method – EDGE (Guo et al., 2021b) to pinpoint the most critical time steps that led the agent to fail its task (See Fig. 1). For a backdoored agent, these steps are most likely to have the trigger presented in the environment. Then, we design a more fine-grained feature-level explanation method to further interpret why the agent took certain actions at the pinpointed steps. As detailed in Section 3.2, our method will identify a subset of features in the state vector that contributes most to the agent's actions at the pinpointed steps. Since the agent's inappropriate actions are mostly driven by the trigger, the identified features are likely to represent the trigger.

Then, we design a backdoor shielding technique to protect the agent from being compromised by the identified trigger. Our high-level idea is to retrain the shielding agent such that it learns to take proper actions at the operating states and keeps its original behaviors at the clean states. As detailed in Section 3.3, we reconstruct the environment with the identified trigger and model this shielding process as a novel optimization problem in the constructed environment. By carefully designing the

policy learning objective function, we could theoretically guarantee that the shielding agent achieves a higher reward in an operating environment while keeping its performance in the pristine clean environment. We leverage the PPO algorithm (Schulman et al., 2017) to update the policy, whose monotonicity property speeds up the convergence and thus improves the retraining efficiency.

Note that we do not assume the knowledge of whether the shielding agent is backdoored or not. In other words, we apply the same detection and shielding process for arbitrary agents. As we will show later in Section 4, by adding proper constraints to the feature-level explanation and shielding retraining, our proposed technique will not affect a clean agent's performance.

## 3.2 TRIGGER RESTORATION

We first run the shielding agent's policy $\pi$ in the operating environment and collect a set of $N$ trajectories $\{\boldsymbol{X}^{(i)}, y_i\}_{i=1:N}$. $\boldsymbol{X}^{(i)} = \{\boldsymbol{s}_t^{(i)}, \boldsymbol{a}_t^{(i)}\}_{t=1:T}$ represents the $i$-th trajectory, where $\boldsymbol{s}_t^{(i)}$ and $\boldsymbol{a}_t^{(i)}$ are the state and action at the time step $t$. $y_i$ denotes the total reward of the $i$-th trajectory. Then, we conduct a step-level explanation to pinpoint the time steps when the trigger is likely to show up. Finally, we conduct a feature-level explanation to identify the trigger at the pinpointed time steps.

**Step-level Explanation.** We leverage a state-of-the-art explanation method EDGE (Guo et al., 2021b). At a high level, EDGE designs a self-explainable model to fit the collected trajectories and thus pinpoint the important time steps in each trajectory. Technically, EDGE proposes the following deep Gaussian process-based model to fit the trajectories.

$$\boldsymbol{f}|\boldsymbol{X} \sim \mathcal{N}(\boldsymbol{0}, \beta_t^2 \boldsymbol{K}_{XX}^t + \beta_e^2 \boldsymbol{K}_{XX}^e)\,, \; y_i|\boldsymbol{F}^{(i)} \sim \begin{cases} \mathrm{Cat}(\mathrm{softmax}(\boldsymbol{f}^{(i)}(\boldsymbol{w}^{(i)})^T)), & \text{If discrete reward} \\ \mathcal{N}(\boldsymbol{f}^{(i)}(\boldsymbol{w}^{(i)})^T, \sigma^2), & \text{otherwise} \end{cases}.$$
(1)

Where $\boldsymbol{f}$ is the output of the deep Gaussian process (DGP) encoder. This DGP encoder first inputs a trajectory $\boldsymbol{X}^{(i)}$ into an RNN encoder and a shallow MLP encoder to obtain an embedding of the state at each time step $\{\boldsymbol{h}_t^{(i)}\}_{t=1:T}$ and an embedding of the whole trajectory $\boldsymbol{e}^{(i)}$. It then designs an additive GP with square exponential (SE) kernel $k_{\gamma_t}$ and $k_{\gamma_e}$ to transform $\boldsymbol{h}$ and $\boldsymbol{e}$ into $\boldsymbol{f} \in \mathbb{R}^{T \times 1}$. With the representation $\boldsymbol{f}_{1:T}^{(i)}$, EDGE then designs a regression model to predict the final reward $y_i$

Since the trigger may present at different time steps in different trajectories, we apply a trajectory-specific mixing weight $\boldsymbol{w}_t^{(i)} = g(\boldsymbol{h}_t^{(i)}, \boldsymbol{e}^{(i)})$, where $g$ is a shallow MLP network. Following Alvarez-Melis & Jaakkola (2018), we add a local linear constrain $\mathcal{L}_e$ to $\boldsymbol{w}^{(i)}$ to ensure its local linearity. By ranking the mixing weight and selecting the time steps associated with the top mixing weights, we can pinpoint the most critical steps for each trajectory. The critical time steps of the failing trajectories are when the agent took inappropriate actions and are thus the most likely to have the trigger presented.

**Feature-level Explanation.** After pinpointing the potential time steps that contain the trigger (denoted as trigger-presented time steps), we then design a feature-level explanation to interpret the agent's actions at those time steps. Given that the trigger causes the agent's inappropriate actions at the trigger-presented time steps. By extracting the key sub-region/features in the state representations at the trigger-presented time steps, we could identify the backdoor trigger. Specifically, we first extract the shielding agent's states and actions at the trigger-presented time steps, denoted as $\mathcal{D} = \{\boldsymbol{s}_t^{(i)}, \boldsymbol{a}_t^{(i)}\}$. Then, we design a feature explanation mask $\boldsymbol{m}$ with the same dimensionality as the state representation $\boldsymbol{s}$. Each element in the mask $\boldsymbol{m}_j$ equals to either 0 or 1, where $\boldsymbol{m}_j = 1$ means the corresponding ($j$-th) element in the state representation $\boldsymbol{s}_t^{(i)}$ is important to the agent's current action $\boldsymbol{a}_t^{(i)}$, otherwise $\boldsymbol{m}_j = 0$. We design $\boldsymbol{m}_j$ to follow a Bernoulli distribution $\mathrm{Bern}(\theta_j)$, with the parameter $\theta_j$. Finally, we add this explanation mask on top of each state in $\mathcal{D}$, input the masked state $\tilde{\boldsymbol{s}}_t^{(i)}$ into the agent's policy network, and obtain a masked action $\tilde{\boldsymbol{a}}_t^{(i)}$.

$$\boldsymbol{m} = \prod_j \mathrm{Bern}(\theta_j)\,, \; \tilde{\boldsymbol{s}}_t^{(i)} = \boldsymbol{s}_t^{(i)} \odot \boldsymbol{m}\,, \; \tilde{\boldsymbol{a}}_t^{(i)} \sim \pi(\tilde{\boldsymbol{s}}_t^{(i)})\,.$$
(2)

Our goal is to mask as many elements in a state $\boldsymbol{s}_t^{(i)}$ as possible but keeping its corresponding masked action $\tilde{\boldsymbol{a}}_t^{(i)}$ as similar as the agent's original action $\boldsymbol{a}_t^{(i)}$. This filters out features that are not important to the agent's current action and only preserves the important ones. As discussed above, the preserved

features most likely represent the trigger that is the reason for the agent's inappropriate actions at the trigger-presented time steps. Note that the trigger of adversarial agent attacks is an action sequence, which is encoded in certain dimensions of the state vector. By highlighting the dimensions pertaining to the adversarial agent's action, our method could identify the trigger actions.

**Explanation Parameter Learning.** We follow Guo et al. (2021b) to solve the step-level explanation model. As for the feature-level explanation model (Eqn. 2), our goal is to minimize the difference between the masked action $\tilde{a}_t^{(i)}$ and the shielding agent's original action $a_t^{(i)}$ at the states in $\mathcal{D}$. This is equivalent to maximize the marginal likelihood $p(a|s, \theta_{1:J})$, which minimizes the difference between the distribution where $\tilde{a}_t^{(i)}$ is sampled from (i.e., $\pi(\tilde{s}_t^{(i)})$) and the agent's original action distribution. Unfortunately, $\log p(a|s)$ is intractable because $m$ is discrete and non-differentiable. To tackle this challenge, we leverage the concrete distribution (Maddison et al., 2016) and Jensen's Inequality to derive a lower bound of $\log p(a|s)$.

**Theorem 1.** *Given $m_j \approx h_\theta(u) = \sigma\big(\frac{\log\alpha_j + \log(u_j/(1-u_j))}{\lambda}\big)$, where $\sigma(\cdot)$ is the sigmoid function, $u_j \sim uniform(0,1)$, and $\alpha_j = \frac{\theta_j}{1-\theta_j}$. We have the following inequality.*

$$\log p(a|s, \theta) \geq \mathbb{E}_u[\log p(a|s, h_\theta(u))]. \tag{3}$$

Appendix A specifies the derivative of Eqn. 3. Instead of maximizing the original log marginal likelihood, we maximize its lower bound. We consider the typical cases where the agent's policy follows a Gaussian or Categorical distribution. Specifically, we have $\log p(a_t^{(i)}|s_t^{(i)}, h_\theta(u))$ equals to $\|a_t^{(i)} - \pi(s_t^{(i)} \odot h_\theta(u))\|$ for the Gaussian cases, and $a\log p(\pi(s_t^{(i)} \odot h_\theta(u) = a)$ for the categorical distribution. With the approximations above, we can solve $\theta_{1:J}$ by maximizing $\mathbb{E}_\mathcal{D}[\mathbb{E}_u[\log p(a_t^{(i)}|s_t^{(i)}, h_\theta(u))]] + \lambda\mathcal{R}(\theta)$ using a first-order optimization method. Here, $\mathcal{R}(\theta)$ is an elastic net (Zou & Hastie, 2005) regularization.

After conducting the two-step explanation, we identify a potential trigger, denoted as $\mathcal{T}$. In particular, for the perturbation-based attack, we compute the average values of the state features selected by $m$ across trigger-present time steps as the trigger patch (Fig. 1). For adversarial agent attacks, we first identify a continuous trigger-present time slice in each trajectory $(t_1, ..., t_L)$. Then, we compute the average value of the selected features in each state across all trajectories $\bar{s}_{t_l} = \frac{1}{N}\sum \tilde{s}_{t_l}^{(i)}$ and use this average state sequence $(\bar{s}_{t_1}, ..., \bar{s}_{t_L})$ as the indicator of the trigger actions.

### 3.3 BACKDOOR SHIELDING

We define two Markov decision processes (MDP) for the shielding agent, $\mathcal{M} = \{\mathcal{S}, \mathcal{A}, \mathcal{R}, \mathcal{P}, \gamma\}$ and $\hat{\mathcal{M}} = \{\hat{\mathcal{S}}, \mathcal{A}, \mathcal{R}, \mathcal{P}, \gamma\}$, where $\mathcal{M}$ and $\hat{\mathcal{M}}$ refers to the MDP of the clean and operating environment. $\hat{s} \in \hat{\mathcal{S}}$ can be poisoned or clean, and we are not aware of its cleanliness. For a multi-agent environment, we fix the policy of the non-shielding agents, and the environment becomes an MDP (Guo et al., 2021a). Given an agent's policy $\pi$, we define its state occupancy distribution in the clean and operating environment as $\rho^\pi(s) = (1-\gamma)\sum_t \gamma^t p(s_t = s|\pi)$ and $\hat{\rho}^\pi(\hat{s}) = (1-\gamma)\sum_t \gamma^t p(\hat{s}_t = \hat{s}|\pi)$. Similarly, we also define the expected total reward of the agent in the clean and operating environment as $\eta(\pi) = \mathbb{E}[\sum_t \gamma^t R(s, \pi(s))]$ and $\hat{\eta}(\pi) = \mathbb{E}[\sum_t \gamma^t R(\hat{s}, \pi(\hat{s}))]$.

**Retraining Objective Function.** Recall that the retraining goal is to improve the agent's performance in the operating environment and maintain its performance in the clean environment. This can be interpreted as the following objective function: $\arg\max_{\hat{\pi}} \hat{\eta}(\hat{\pi})$, s.t. $|\eta(\hat{\pi}) - \eta(\pi)| \leq \epsilon$, where $\hat{\pi}$ is the retrained policy. To solve this objective function, we first define the following approximation of $\hat{\eta}(\hat{\pi})$ based on $\hat{\eta}(\pi)$. $L_\pi(\hat{\pi}) = \hat{\eta}(\pi) + \sum_{\hat{s}} \hat{\rho}^\pi(\hat{s})\sum_a \hat{\pi}(a|\hat{s})A_\pi(\hat{s}, a)$, where $A$ is the advantage function. According to Schulman et al. (2015), we have the following inequality $\eta(\hat{\pi}) \geq M_\pi(\hat{\pi}) = [L_\pi(\hat{\pi}) - C\max_{\hat{s}\sim\hat{\rho}^\pi}\mathbb{KL}(\pi(\cdot | \hat{s})\|\hat{\pi}(\cdot | \hat{s}))]$. By maximizing $M_\pi(\hat{\pi})$, we can guarantee that $M_\pi(\hat{\pi}) \geq M_\pi(\pi) = \hat{\eta}(\pi)$. As a result, without considering the constraint, we can guarantee the performance of the retrained agent will be improved in the operating environment, i.e., $\hat{\eta}(\hat{\pi}) \geq \hat{\eta}(\pi)$.

To realize the constraint $|\eta(\hat{\pi}) - \eta(\pi)| \leq \epsilon$, we first introduce the following theorem.

**Theorem 2.** *Given a policy $\pi$ and its retrained policy $\hat{\pi}$, we have the following inequality $|\eta(\pi) - \eta(\hat{\pi})| \leq C\max_{s\sim\rho^\pi}\mathbb{KL}(\pi(\cdot | s)\|\hat{\pi}(\cdot | s))$.*

See Appendix A for the proof. Theorem 2 states that by constraining the maximum KL divergence between $\pi$ and $\hat{\pi}$ in the clean states, we can bound the difference between the $\eta(\pi)$ and $\eta(\hat{\pi})$ and thus maintain the agent's performance in the clean environment.

As such, we can transform the objective function above to the following objective function.

$$\operatorname*{argmax}_{\hat{\pi}} L_{\pi}(\hat{\pi}) \, , \text{ s.t. } \begin{cases} \hat{K} = \mathbb{E}_{\hat{s} \sim \hat{\rho}(\pi)}[\mathbb{KL}(\pi(\cdot \mid \hat{s}) \| \hat{\pi}(\cdot \mid \hat{s}))] \leq \epsilon \, , \\ K = \mathbb{E}_{s \sim \rho(\pi)}[\mathbb{KL}(\pi(\cdot \mid s) \| \hat{\pi}(\cdot \mid s))] \leq \epsilon \, . \end{cases} \tag{4}$$

By solving Eqn. 4, we theoretically guarantee that the shielding agent achieves a higher reward in an operating environment ($\hat{\eta}(\hat{\pi}) \geq \hat{\eta}(\pi)$) while keeping its performance in the clean environment ($|\eta(\hat{\pi}) - \eta(\pi)| \leq \epsilon$).

**Retraining Algorithm.** We conduct the retraining in the operating environment. To approximate the constraint $K$ in Eqn. 4, we need to identify a set of clean states in the operating environment. For the perturbation-based attack, we apply the explanation mask to the current state and compare the feature values in the highlighted region with $\mathcal{T}$. If their difference is within a certain threshold, we deem $s_t$ as a poisoned state; otherwise, we treat it as a clean state. We apply similar operations for the adversarial agent attack to compare the agent's current action with the identified trigger actions. Algorithm 1 in Appendix A shows our final backdoor shielding algorithm. At a high level, we approximate $K$ with the clean states identified from the operating environment and retrain the shielding agent by maximizing Eqn. 4. Appendix B specifies implementation and hyper-parameters.

# 4 EVALUATION

## 4.1 EXPERIMENT SETUP

**Attacks and Environments.** We first select Trojdrl (Kiourti et al., 2019), a perturbation-based attack against single-player environments. Trojdrl adds a $3 \times 3$ square patch to the agent's state representation and forces the backdoored agent to take either a pre-specified action (targeted attack) or a random action (untargeted attack) at poisoned states. Here, we mainly test SHINE against the targeted, and Appendix E shows the results against the untargeted attack. We follow Trojdrl and select three Atari games from the OpenAI Gym (Brockman et al., 2016) environment pool - Pong, Breakout, and Space Invaders. We also consider a perturbation-based attack against the multi-agent cooperative RL (Chen et al., 2022b). We select the SMAC environment and use the default attack setup to launch two attacks: one against a Q-learning algorithm QMIX and the other against a policy gradient algorithm COMA. Appendix A discusses how to adapt SHINE to this attack against multi-agent RL. Regarding the adversarial agent attack, we select Backdoorl (Wang et al., 2021a), designed for two-player competitive Markov games. We also use three MuJoCo environments selected by Backdoorl, i.e., You-Shall-Not-Pass, Sumo-Humans, and Run-To-GO-Ants (Todorov et al., 2012). We follow Backdoorl and set the trigger actions as standing still for a few time steps.

Using the selected environments, we first train a clean agent to achieve near-optimal performance. Then, we train backdoored agents such that each agent's reward reduces dramatically when the trigger is presented while keeping near-optimal in the clean environment. Finally, we simulate the operating environment by presenting the trigger in the environment with probability $P_{\alpha}$ at each time step.

**Baseline.** There is no existing work that considers the same setup and defense goal as ours. We first select two state-of-the-art trigger restoration techniques designed for classifiers – NC (Wang et al., 2019) and FeatureRE (Wang et al., 2022). In particular, we collect a set of states and actions of the shielding agent and run these methods by treating the agent's actions as the target classes. We deem the restored trigger with the smallest $l_0$ norm as the backdoor trigger. We use our proposed retraining method on the triggers identified by the NC and FeatureRE to retrain the shielding agent. Note that these methods cannot be applied to adversarial agent attacks or environments with a continuous action space. We also consider a straightforward baseline, that is, to directly retrain the shielding agent in the operating environment using the PPO algorithm without applying any shielding (denoted as "Direct retraining"). Section 2 discusses two other existing DRL backdoor defenses. Bharti et al. (2022) requires accessing clean environments. Appendix C shows that SHINE is more effective, generalizable, and scalable than Bharti et al. (2022).[1]

---

[1] Another defense (Guo et al., 2022) is still a pre-print paper without public implementation. Besides, it cannot be applied to perturbation-based attacks. As such, we do not compare it with SHINE in our experiments.

Table 1: Trigger restoration fidelity of SHINE , NC, and FeatureRE. Note that "-" means not applicable.

| Method | Pong | Breakout | Space Invaders | SMAC | You-Shall-Not-Pass | Sumo-Humans | Run-to-Go-Ants |
|---|---|---|---|---|---|---|---|
| NC | 0.268±0.033 | 0.296±0.020 | 0.217±0.036 | 0.045±0.009 | - | - | - |
| FeatureRE | 0.936±0.005 | 0.962±0.012 | 0.927±0.006 | 0.896±0.011 | - | - | - |
| SHINE | **0.998**±0.001 | **0.997**±0.001 | **0.998**±0.001 | **0.936**±0.003 | **0.930**±0.002 | **0.973**±0.006 | **0.981**±0.003 |

Table 2: Performance of *backdoored* agents retrained with different methods in the operating and clean environment. We report the average score or average winning rate across 1,000 game rounds. We also conduct a paired t-test to demonstrate the statistical significance of results in this table and analyze the agent's action distribution in clean and poisoned states of the operating environment. The results are shown in Appendix F.

| Environment | Method | Pong | Breakout | Space Invaders | QMIX (%) | COMA (%) | You-Shall-Not-Pass (%) | Sumo-Humans (%) | Run-to-Go-Ants (%) |
|---|---|---|---|---|---|---|---|---|---|
| Operating | Original | -0.010±0.001 | 16.40±0.48 | 108.3±2.9 | 19.6±0.1 | 62.0±0.7 | 17.9±0.4 | 10.4±0.2 | 16.2±0.2 |
| | Direct retraining | 0.032±0.002 | 20.37±1.72 | 639.0±5.4 | 82.6±0.5 | 88.9±1.8 | 27.8±4.3 | 23.6±2.8 | 37.6±0.3 |
| | NC | -0.102±0.005 | 11.50±0.69 | 293.7±1.4 | 23.8±0.3 | 21.9±0.6 | - | - | - |
| | FeatureRE | 0.124±0.003 | 15.36±1.02 | 403.3±2.1 | 33.2±0.5 | 23.9±0.1 | - | - | - |
| | SHINE | **0.728**±0.027 | **28.63**±2.05 | **832.9**±7.5 | **99.1**±0.8 | **92.3**±1.2 | **48.2**±1.8 | **32.4**±1.3 | **52.4**±0.6 |
| Clean | Original | 0.680±0.030 | 22.33±1.05 | 685.3±3.5 | 99.6±0.9 | 96.3±0.5 | 49.8±0.2 | 29.3±0.1 | 52.0±0.4 |
| | Direct retraining | 0.286±0.016 | 21.82±1.90 | 723.2±2.7 | 99.1±0.6 | 96.9±1.2 | 38.4±3.2 | 23.5±3.6 | 51.0±1.2 |
| | NC | 0.136±0.013 | 12.66±1.02 | 301.6±2.8 | 99.3±0.1 | 97.3±0.6 | - | - | - |
| | FeatureRE | 0.293±0.021 | 22.57±1.42 | 703.9±9.2 | 99.2±0.1 | 96.9±0.5 | - | - | - |
| | SHINE | 0.734±0.021 | 25.35±1.60 | 835.1±3.6 | 99.2±0.8 | 97.0±0.5 | 49.5±1.6 | 33.5±1.8 | 52.9±0.6 |

## 4.2 EXPERIMENT DESIGN

**Exp-I: Trigger Restoration Faithfulness.** We first evaluate whether SHINE faithfully identifies the trigger. For perturbation-based attacks, we compare the trigger $\mathcal{T}$ identified by SHINE with the real trigger $\hat{\mathcal{T}}$ and compute the precision $\frac{\|\mathcal{T}\odot\hat{\mathcal{T}}\|_1}{\|\mathcal{T}\|_1}$ and recall $\frac{\|\mathcal{T}\odot\hat{\mathcal{T}}\|_1}{\|\hat{\mathcal{T}}\|_1}$. We report the F1 score as the faithfulness metric. We compare SHINE with NC and FeatureRE under the perturbation-based attacks. Since adversarial agent attacks do not have a fixed trigger patch, we compare the trigger actions identified by SHINE with the real ones designed by the attackers and report the F1 score.

**Exp-II: Backdoor Shielding Effectiveness.** Second, we evaluate the efficacy of SHINE in backdoored agent shielding. In particular, we retrain the backdoored agents in Exp-I using our proposed retraining algorithm and report their performance in the operating and clean environment before and after retraining. We compare SHINE with the three baselines mentioned above.

**Exp-III: SHINE on Clean Agents.** We also apply SHINE to the clean agent in each environment to verify that SHINE will not affect a clean agent's performance. Similar to Exp-II, We report each agent's performance in the operating and clean environment before and after shielding.

**Exp-IV: Sensitivity of SHINE against Attack Variations.** We use the Atari-pong environment of the perturbation-based attack to test the sensitivity of SHINE against attacks varied in trigger patterns, sizes, and poison rates. We craft eight attack variations. We first vary the trigger pattern from the dense patch to more incompact triggers (Cross and Equal sign). We also keep the trigger pattern as dense and vary the trigger size ($3 \times 3$, $4 \times 4$, $5 \times 5$) and the trigger presenting probabilities ($P_\alpha = 0.1/0.2/0.3$). In each environment, we train a backdoored agent, shield it with our method and two baseline approaches (Direct training and NC), and report the shielding performance.

We repeat each experiment 3 times with different random seeds and report the mean and standard deviation (std). Furthermore, we conduct an ablation study, demonstrate the computational efficiency, evaluate the hyper-parameter sensitivity, and test SHINE against more attack variations. Due to the space limit, we present these experiments in Appendix D& F.

## 4.3 EXPERIMENT RESULTS

**Results of Exp-I.** Tab. 1 shows the trigger restoration fidelity. Both NC and FeatureRE show limited capability in trigger restoration because these methods do not consider DRL's sequential decision-making nature. In contrast, SHINE can faithfully identify the trigger for both perturbation-based attacks and adversarial agent attacks, verifying the effectiveness of our trigger restoration technique (Appendix F showcases the identified triggers). The ablation study in Appendix F further demonstrates the efficacy of step-level and feature-level explanations, respectively.

Table 3: Performance of *clean* agents retrained with SHINE in the operating and clean environment. We report the average score and average winning rate across 1,000 game rounds, respectively.

| Environment | Method | Pong | Breakout | Space Invaders | QMIX (%) | COMA (%) | You-Shall-Not-Pass (%) | Sumo-Humans (%) | Run-to-Go-Ants (%) |
|---|---|---|---|---|---|---|---|---|---|
| Operating | Original | 0.816±0.040 | 19.50±0.59 | 835.6±1.8 | 99.0±0.2 | 96.3±0.1 | 50.2±0.3 | 34.5±0.2 | 53.5±0.2 |
| | SHINE | 0.818±0.038 | 24.99±1.07 | 838.2±3.6 | 99.3±0.6 | 96.9±1.7 | 51.3±0.9 | 35.3±0.5 | 53.9±0.3 |
| Clean | Original | 0.778±0.067 | 27.50±1.32 | 836.4±3.2 | 99.8±0.1 | 96.6±0.1 | 51.2±0.1 | 35.8±0.2 | 54.5±0.1 |
| | SHINE | 0.769±0.089 | 26.27±0.64 | 838.2±3.4 | 99.6±0.1 | 97.9±0.2 | 51.5±0.5 | 35.7±1.2 | 53.8±0.2 |

**Results of Exp-II and Exp-III.** Tab. 2 shows the performance of backdoored agents shielded by SHINE and three baseline methods. First, NC and FeatureRE have limited efficacy due to the low fidelity of their resolved triggers. Direct retraining could improve the retrained agent's performance in the operating/poisoned environment, but the improvement is still limited. More importantly, due to environmental variations, it cannot preserve the retrained agent's effectiveness in the original clean environment. Note that direct retraining outperforms NC and FeatureRE in some cases, demonstrating the negative impact of retraining with non-faithful triggers. In comparison, benefiting from its high-fidelity trigger, the agent retrained by SHINE achieves the highest performance in the operating environment of all the games. In addition, SHINE well retains (or even improves) the retrained agent's effectiveness in the original clean environment. This result is aligned with our theoretical analysis in Section 3.3, verifying the effectiveness of our backdoor shielding in robustifying a backdoored agent while preserving its generalizability.

Tab. 3 shows a clean agent's performance before and after shielding it with SHINE . As shown in the table, SHINE introduces a minor performance drop or improves the clean agent's performance in both operating and clean environments. This is an important property in that users could directly apply SHINE to arbitrary agents without making critical decisions of which ones are truly backdoored agents, which, in general, is sensitive to the choice of detection threshold.

**Results of Exp-IV.** Fig. 2 shows the retraining performance of SHINE and baseline methods against different attack variations. SHINE only has marginal performance variations in different setups, verifying SHINE's insensitivity against attack variations. This verifies the generalizability and practicability of SHINE in that users could apply SHINE without tailoring for different attack variations. Fig. 2 also demonstrates that SHINE outper-

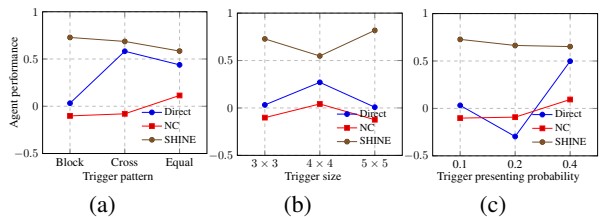

(a)     (b)     (c)

Figure 2: The average score of the agent retrained with different shielding methods against attack variations.

forms baselines against all attack variations, further demonstrating its superiority over these methods. In appendix F, we demonstrate the robustness of SHINE against more variations in trigger shapes and sizes.

## 5 DISCUSSION

**Adaptive Attacks.** We first consider a straightforward adaption of the perturbation-based attack. Specifically, we allow the trigger to present at different locations in the environment snapshot at different time steps. This adaptive attack with a dynamic trigger could potentially bypass our method because our feature-level explanation resolves a fixed trigger mask across all the trigger-presented time steps. We follow the attack method in TrojDRL (Kiourti et al., 2019) and try to launch this attack in the Pong environment. The attack cannot succeed even after we carefully tune the training parameters. We can only reduce the agent's average score in the operating environment by 30.3% and 1.2% for targeted and untargeted attacks, respectively. Our attempts motivate future work to design stronger backdoor attacks that could inject this dynamic trigger. Even if this attack succeeds, we can adjust SHINE to defend against it. First, rather than obtaining a common explanation mask, we first solve a specific mask for each state to capture the trigger movement. During retraining, we slide our identified trigger $\mathcal{T}$ across the whole state representation to decide its cleanliness (Appendix A).

Another possible adaptive attack is to attack our explanation methods. We found only one existing attack (Huai et al., 2020) that targets gradient/saliency-based explanation methods. Since both our step-level and feature-level explanations have a different mechanism from these gradient-based methods, this attack cannot be directly applied to our method. To enable an effective attack against our explanation method, the attacker needs to first design a proper attack against either step-level or feature-level explanation method. The attacker also needs to consider how to properly design this attack, such that it could work together with the trojan attack against the DRL agent. We believe designing such an attack requires non-trivial effort and leave it as our future work.

**Distinction from Environment-perturbation Attacks and Robust RL.** Our problem differs from environment-perturbation attacks (Russo & Proutiere, 2019; Zhang et al., 2021; Liang et al., 2022; Sun et al., 2021; Kamalaruban et al., 2020), which attack a pretrained agent by adding perturbations to its observations. Defenses against such attacks typically limit the perturbation strength within a $\epsilon$-norm ball and design methods to train optimal policies under this perturbation ball. Similarly, some robust RL methods (Pinto et al., 2017; Tessler et al., 2019) train policies under random perturbations to the agent's observations or actions. Due to differences in attack/problem setups, these methods cannot be directly applied to our problem. Our future work will explore novel DRL backdoor defenses by following the idea of these works. For instance, we will investigate how to model the defense against trojan attacks as a Partially Observable MDP and train a robust policy accordingly.

**Postmortem Defense.** SHINE operates in a postmortem fashion, meaning it conducts shielding after a backdoor is triggered. This approach significantly lowers the threshold and cost for defense since it does not require a guaranteed clean environment, which can be challenging to construct. We acknowledge that for extremely critical applications, triggering the attack can be costly. Recall that SHINE can effectively identify triggers even when they occur very infrequently ($P_\alpha = 0.1$). This suggests that SHINE does not require extensive failed cases to apply shielding. For critical applications, SHINE offers a rapid response once an attack occurs, minimizing future damage. It represents a practical defense approach in environments where attackers can access.

**More complicated triggers.** As mentioned in Section 4, we first use default triggers of existing attacks to evaluate our method. Then, we demonstrate the effectiveness of SHINE against perturbation-based attacks with more complicated and incompact triggers (Exp-IV. and Appendix F). Furthermore, existing attacks against supervised classifiers also explore other trigger patterns (e.g., watermarks), which have not been used for DRL attacks yet. In future work, we will explore designing effective attacks with such trigger patterns and extending SHINE to defend against these attacks.

**Limitations.** First, existing research proposes other methods to explain a DRL agent's action (e.g., Atrey et al. (2019); Greydanus et al. (2018); Puri et al. (2020)). Our future research endeavor will explore extending these methods to identify the backdoor trigger in our context. Second, we compare SHINE with NC and FeatureRE, two state-of-the-art backdoor detection methods for supervised classifiers. While recent research proposes other methods Wang et al. (2023); Tao et al. (2022), they are still designed for supervised classifiers and suffer similar limitations. Thus, they have limited efficacy and generalizability in our problem. Our future work will test whether these methods can yield better results than NC against perturbation-based attacks in environments with a discrete action space. Third, we evaluate SHINE against the attacks in three types of environments. Our future work will extend the backdoor attacks and defenses to broader types of games, including extensive-form games (Go (Tian et al., 2019), board games (Lanctot et al., 2019)), multi-agent competitive games (Zhang et al., 2019). Finally, we show that SHINE can still detect and shield a backdoored agent even when the trigger varies in size and presenting timesteps/probability. Our future work will explore providing a theoretical guarantee against these variations.

## 6 CONCLUSION

This work proposes SHINE, a method for shielding DRL agents against backdoor attacks. SHINE first identifies the backdoor trigger presented in the environment and then retrains the DRL agent to eliminate the influence of the trigger on its policy. Our experiments in various benchmark RL environments demonstrate SHINE's efficacy in shielding backdoored agents against different backdoor attacks while maintaining the clean agents' performance. With all these experiments and analyses, we safely conclude that through explanation and retraining, we can effectively shield DRL agents from backdoor attacks in a practical scenario.

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

# A ADDITIONAL TECHNICAL DETAILS

## A.1 PROOF OF THEOREM 1.

Based on Jensen's inequality, we have the following derivative of the log marginal likelihood $\log p(\boldsymbol{a}|\boldsymbol{s}, \theta)$.

$$\log p(\boldsymbol{a}|\boldsymbol{s}, \theta) = \log \int p(\boldsymbol{a}, \boldsymbol{m}|\boldsymbol{s}, \theta)\mathrm{d}\boldsymbol{m} = \log \int p(\boldsymbol{a}|\boldsymbol{m}, \boldsymbol{s}, \theta)p(\boldsymbol{m}|\theta)\mathrm{d}\boldsymbol{m}$$
$$= \log \mathbb{E}_{\boldsymbol{m}}[p(\boldsymbol{a}|\boldsymbol{m}, \boldsymbol{s}, \theta)] \geq \mathbb{E}_{\boldsymbol{m}}[\log p(\boldsymbol{a}|\boldsymbol{m}, \boldsymbol{s}, \theta)].$$
(5)

As stated in Theorem 1, by defining $\boldsymbol{u}$, where $\boldsymbol{u}_j \sim \text{uniform}(0,1)$, and $\alpha_j = \frac{\theta_j}{1-\theta_j}$, we can approximate $\boldsymbol{m}_j$ with $h_\theta(\boldsymbol{u}) = \sigma(\frac{\log\alpha_j + \log(\boldsymbol{u}_j/(1-\boldsymbol{u}_j))}{\lambda})$, where $\sigma(\cdot)$ is the sigmoid function. With $\boldsymbol{u}$, we can then derive the following inequality from Eqn. equation 5

$$\log p(\boldsymbol{a}|\boldsymbol{s}, \theta) \geq \mathbb{E}_{\boldsymbol{m}}[\log p(\boldsymbol{a}|\boldsymbol{m}, \boldsymbol{s}, \theta)] \approx \mathbb{E}_{\boldsymbol{u}}[\log p(\boldsymbol{a}|\boldsymbol{s}, h_\theta(\boldsymbol{u}))].$$
(6)

$\square$

## A.2 PROOF OF THEOREM 2.

We define $H_\pi(\hat{\pi}) = \eta(\pi) + N_\pi(\hat{\pi}) = \eta(\pi) + \sum_{\boldsymbol{s}} \rho^\pi \sum_{\boldsymbol{a}} \hat{\pi}(\boldsymbol{a} \mid \boldsymbol{s})A_\pi(\boldsymbol{s}, \boldsymbol{a})$. According to Schulman et al. (2015), we have

$$|\eta(\hat{\pi}) - H_\pi(\hat{\pi})| \leq C_1 \max_{\boldsymbol{s} \sim \rho^\pi} \mathbb{KL}(\pi(\cdot|\boldsymbol{s})||\hat{\pi}(\cdot|\boldsymbol{s}))$$
$$|\eta(\hat{\pi}) - \eta(\pi) - N_\pi(\hat{\pi})| \leq C_1 \max_{\boldsymbol{s} \sim \rho^\pi} \mathbb{KL}(\pi(\cdot|\boldsymbol{s})||\hat{\pi}(\cdot|\boldsymbol{s}))$$
$$|\eta(\hat{\pi}) - \eta(\pi)| \leq C_1 \max_{\boldsymbol{s} \sim \rho^\pi} \mathbb{KL}(\pi(\cdot|\boldsymbol{s})||\hat{\pi}(\cdot|\boldsymbol{s})) + |N_\pi(\hat{\pi})|.$$
(7)

According to Theorem 1 in Achiam et al. (2017), we can derive

$$|\sum_{\boldsymbol{s}} \rho^\pi(\boldsymbol{s}) \sum_a \hat{\pi}(\boldsymbol{a} \mid \boldsymbol{s})A_\pi(\boldsymbol{s}, \boldsymbol{a})| \leq \mathbb{E}_{\boldsymbol{s} \sim \rho, \boldsymbol{a} \sim \pi, \boldsymbol{s}' \sim p(\cdot|\boldsymbol{s}, \boldsymbol{a})}[(\frac{\hat{\pi}(\boldsymbol{a}|\boldsymbol{s})}{\pi(\boldsymbol{a}|\boldsymbol{s})} - 1)R(\boldsymbol{s}, \boldsymbol{a}, \boldsymbol{s}')]$$
$$= \mathbb{E}_{\boldsymbol{s} \sim \rho, \boldsymbol{a} \sim \pi}[(\hat{\pi}(\boldsymbol{a}|\boldsymbol{s}) - \pi(\boldsymbol{a}|\boldsymbol{s}))\mathbb{E}_{\boldsymbol{s}' \sim p(\cdot|\boldsymbol{s}, \boldsymbol{a})}R(\boldsymbol{s}, \boldsymbol{a}, \boldsymbol{s}')]$$
$$\leq C_2 \max_{\boldsymbol{s} \sim \rho^\pi} \mathbb{KL}(\pi(\cdot|\boldsymbol{s})||\hat{\pi}(\cdot|\boldsymbol{s})).$$
(8)

Based on Eqn. equation 7 and equation 8, we have the following inequality.

$$|\eta(\pi) - \eta(\hat{\pi})| \leq C \max_{\boldsymbol{s} \sim \rho^\pi} \mathbb{KL}(\pi(\cdot \mid \boldsymbol{s})||\hat{\pi}(\cdot \mid \boldsymbol{s}))$$
(9)

$\square$

## A.3 BACKDOOR SHIELDING ALGORITHM.

Algorithm 1 shows our final backdoor shielding algorithm. Note that, for perturbation-based attacks, to further filter out false positive triggers, we leverage the assumption that a trigger is small and visually imperceptible and add a trigger filter before using the identified trigger for retraining. Specifically, we compute the $l_0$-norm of $\mathcal{T}$ and only use it if $\|\mathcal{T}\|_0$ is smaller than a threshold (e.g., 5% of the whole state representation features).

## A.4 ADAPTIONS OF SHINE.

**Adaption for Multi-agent Attacks.** Recall that we also apply SHINE against an existing attack for multi-agent cooperative RL in the SMAC environment. We use in-distribution triggers with a trigger size of 5%. We use the 2s3z and 3m map for QMIX and COMA, respectively. To run our method, we need to apply the following adoptions. Specifically, we first applied our explanation-based trigger detection method on the local observations of the cooperative agents to identify the backdoor trigger. In particular, for QMIX, we performed the explanation for each local Q function and then aggregated the identified triggers. For COMA, we directly used the global Q function since it takes the central states as input. Using the identified trigger, we then applied the shielding procedure. Since this attack

---

**Algorithm 1** Backdoor shielding algorithm.

---

1: **Input:** the operating environment that the trigger show ups at each state with a certain probability, the shielding agent's original policy $\pi$, the identified trigger $\mathcal{T}$, threshold $\eta_1$ and $\eta_2$, retraining iteration $L$
2: **for** $l = 1$ to $L$ **do**
3:     Run the current policy $\hat{\pi}^{(l-1)}$ in the environment and collect a set of trajectories $\mathcal{I}$ .
4:     Define a clean state set $\mathcal{C}$.
5:     **for** $i = 1$ to $|\mathcal{I}|$ and $t = 1$ to $T$ **do**
6:         **if** $\|s_t^i \odot m - \mathcal{T}\| \geq \eta_2$ **then**
7:             $\mathcal{C} = \mathcal{C} \cup s_t^i$
8:         **end if**
9:     **end for**
10:     Approximate $K$ in Eqn. (5) with $\tilde{K} = \mathbb{E}_{s \in \mathcal{C}}[\mathbb{KL}(\pi(\cdot \mid s) \| \hat{\pi}^{(l-1)}(\cdot \mid s))]$
11:     Plug $\tilde{K}$ in Eqn. (5),
12:     Using $\mathcal{I}$, which contains poisoned states to compute $\hat{K}$ and $L_\pi(\hat{\pi})$.
13:     Update $\hat{\pi}^{(l-1)}$ by solving Eqn. (5) and obtain the updated policy $\hat{\pi}^{(l)}$
14: **end for**
15: Return the retrained policy: $\hat{\pi} = \hat{\pi}^{(L)}$

---

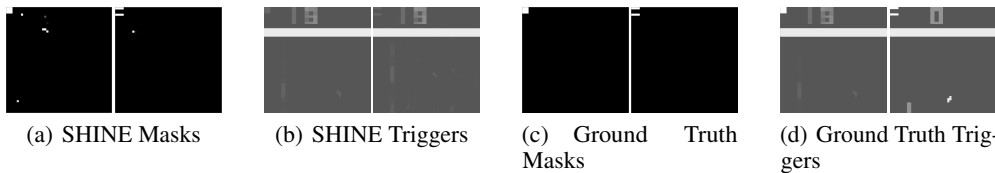

   (a) SHINE Masks       (b) SHINE Triggers      (c) Ground Truth Masks      (d) Ground Truth Triggers

Figure 3: Triggers and masks solved by SHINE . We only show $50 \times 50$ pixels in the top-left corner of the states for better visualization purposes, since the trigger size is small.

considers multiple victim agents, we retrain each agent using algorithm 1. When updating the policy (line 13), we leverage the agent's original training method (i.e., either QMIX or COMA).

**Adaption for Adaptive Attacks.** As mentioned in Section 4, to defend against the adaptive attack where the trigger varies its location across time, we also need to adapt our defense to solve a specific mask for each state. Specifically, we select the states of the top important time steps identified by our step-level explanation. These states are those where the trigger is most likely to be presented. For each state, we perturb its representation by adding Gaussian noise. Then, we use the original and perturbed states as the input data and apply our feature-level explanation to solve an explanation mask. We solve an explanation mask for each of the selected states, which highlights the different subset of features in the state representation (i.e., different locations in the environment), capturing the movement of the trigger. By applying the explanation masks solved from the select states to their representations, we could identify the common trigger pattern.

Note that this adaption (trigger location variation) is not applicable to the adversarial-agent attack, which does not have a perturbation patch as the trigger. Besides, in the MuJoCo games, the feature meanings in the state representation are pre-defined. In other words, specific dimensions of the state representation correspond to the adversarial agent's status, including its position and action. This indicates that no matter how the adversarial agent changes its trigger action, it will pertain to fixed dimensions in the state representation, giving a similar explanation mask.

## B   Implementation and Hyper-parameters

**Implementations.** We use the pytorch (Paszke et al., 2019) and the gpytorch (Gardner et al., 2018) package to implement the trigger detection step of SHINE and stable-baseline (Raffin et al., 2019) to implement the backdoor shielding of SHINE.

Table 4: The average score of the agent shielded by SHINE in the operating and clean environment of the Pong game.

Table 5: The average score of backdoored agents shielded with different methods in the operating and clean environment. The agents are subject to untargeted attacks.

| Environment | $\lambda$ | | | $\epsilon$ | | |
|---|---|---|---|---|---|---|
| | $1 \times 10^{-4}$ | $2 \times 10^{-4}$ | $3 \times 10^{-4}$ | 0.005 | 0.01 | 0.05 |
| Operating | 0.728 | 0.784 | 0.644 | 0.765 | 0.728 | 0.714 |
| Clean | 0.734 | 0.732 | 0.522 | 0.734 | 0.734 | 0.333 |

| Policy | Operating Environment | Clean Environment |
|---|---|---|
| Direct retraining | 0.412 | 0.560 |
| NC | 0.210 | 0.160 |
| SHINE | **0.530** | **0.574** |

**Hyper-parameters.** The key hyper-parameters introduced by our method are the weight of the elastic-net regularization term in the feature-level explanation $\lambda$ and the strength of the KL constraint in the policy retraining $\epsilon$. We set $\lambda$ to $1 \times 10^{-4}$ and $\epsilon$ to 0.01. In addition, our method inherits hyperparameters from the selected step-level explanation method – EDGE (Guo et al., 2021b), the policy updating method – PPO (Schulman et al., 2017), and the temperature in the concrete distribution (Maddison et al., 2016). For those hyper-parameters, we use the default choices in their original implementations.

## C  SHINE AGAINST LATEST BACKDOOR DEFENSE IN RL

We also evaluate SHINE against the latest backdoor defense against the perturbation-based attack (Bharti et al., 2022). More specifically, we relax our assumption of only accessing to the agent's operation environment. Instead, we allow access to the original clean environment and apply the defense in Bharti et al. (2022). We compare the backdoored agent's performance after shielding with our method and the method in Bharti et al. (2022) using the Breakout environment (which is originally used by Bharti et al. (2022) to evaluate their method). Note that we use the same backdoored agent in our evaluation, and shield the agent with SHINE under the same environment. We use their official implementation and adopt the default hyperparameters.

The average score of the agent shielded by our method and Bharti et al. (2022) across 1,000 rounds is 28.63 and 2.5, respectively. SHINE achieves better performance than Bharti et al. (2022), verifying its effectiveness. We note that the results are at a different scale from the reported numbers in Bharti et al. (2022). This is because we set a maximum length of 2,000 for each game round, while they run the game for an unlimited time until it stops naturally. As such, the agents in their experiments collect more scores than those in our setup. Note that, following the suggestion in Bharti et al. (2022), we tried multiple attempts for the singular value thresholds and reported the best result.

In addition, Bharti et al. (2022) requires computing the eigen-decomposition of the concatenation of the state representation, whose complexity is cubic to the dimensionality of the input matrix. This indicates the method in Bharti et al. (2022) will encounter scalability issues when handling environments whose state representation is of high dimensionality and the trajectory is long. Besides, this method is designed only for perturbation-based attacks. As such, SHINE is more generalizable and scalable than the method in Bharti et al. (2022).

## D  COMPUTATIONAL EFFICIENCY AND HYPER-PARAMETER SENSITIVITY

**Runtime.** On average, the trigger detection stage of SHINE takes 12 hours, and the retraining stage takes 5 hours on a single NVIDIA RTX A6000 GPU. We believe this runtime is reasonable in that it is still within the normal range of training a DRL agent in benchmark environments.

**Hyper-parameter Sensitivity.** Recall that SHINE introduces two unique hyper-parameters – the elastic-net regularization term in the feature-level explanation $\lambda$ and the strength of the KL constraint in the policy retraining $\epsilon$. Here, we vary these parameters and observe their influence on SHINE's performance. In particular, we conduct the experiment on the Pong environment, with the trigger setup the same as the Exp-I in Section 5. We evaluate three choices of $\lambda$: $1 \times 10^{-4}$, $2 \times 10^{-4}$, $3 \times 10^{-4}$ and three choices of $\epsilon$: $0.005, 0.01, 0.05$. Tab. 4 shows the performance of the shielded agent in the operating and clean environment under different hyper-parameter settings. The results show that SHINE is insensitive to the subtle variations in these two hyper-parameters.

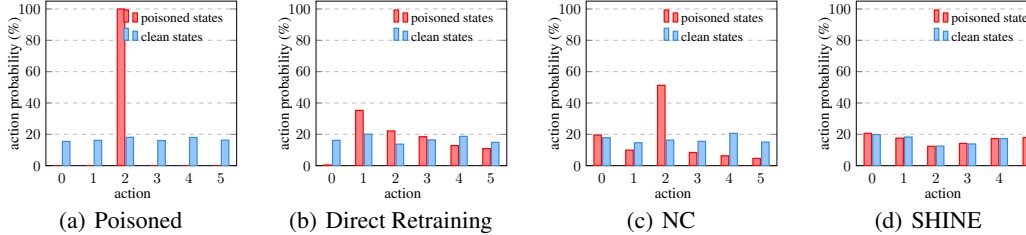

Figure 4: Action distributions in poisoned and clean states in the operating environment of the Atari-Pong Game, corresponding to the results of Exp-II. A poisoned policy exhibits a higher probability of taking the target action (#2) in poisoned states than in clean states.

Table 6: Paired t-test p-value between the performance of backdoored agents retrained with our method and those retrained with the baseline methods in the operating environment.

| Method | Pong | Breakout | Space Invaders | QMIX | COMA | You-Shall-Not-Pass | Sumo-Humans | Run-to-Go-Ants |
|---|---|---|---|---|---|---|---|---|
| Original | >0.001 | >0.001 | >0.001 | >0.001 | >0.001 | >0.001 | >0.001 | >0.001 |
| Direct retraining | >0.001 | 0.013 | >0.001 | 0.001 | 0.004 | 0.002 | 0.004 | >0.001 |
| NC | >0.001 | >0.001 | >0.001 | >0.001 | >0.001 | - | - | - |
| FeatureRE | >0.001 | >0.001 | >0.001 | >0.001 | >0.001 | - | - | - |

# E    SHINE AGAINST UNTARGETED ATTACK IN TROJDRL.

In Tab. 5, we show the average score of SHINE and baselines on the Pong game against the untargeted attack in TrojDRL (with the same trigger as the Exp-I. in Section 5). The table shows that SHINE is still more effective than the selected baseline methods against the untargeted attack, which further demonstrates the effectiveness of our method.

# F    OTHER EXPERIMENTS.

**Paired t-test for Results in Tab.2.** We conduct a paired t-test to demonstrate the statistical significance of our comparison results in Tab. 2. More specifically, our null hypothesis is $H_0 : \mathbb{E}[D] \leq 0$, where $D$ is the reward difference between our method and a baseline method. If the $p$-value is larger than an empirical threshold (e.g., 0.05), we accept $H_0$, indicating our method cannot outperform the baseline. We report the results in Tab. 6.

**SHINE against more Trigger Variations.** Using the same pong environment as Exp-IV, we consider three trigger shapes: dense square block, cross sign, and equal sign. For each shape, we consider four different sizes: $3 \times 3$, $4 \times 4$, $5 \times 5$, and $6 \times 6$. We use these 16 triggers to launch the TrojDRL attack and run SHINE for defending. We report the trigger fidelity and final shielding performance in the operating environment in Tab. 7 and Tab. 8. The results show that our method, including the feature-level explanation, is robust against these variations. Note that, for all the variations, we run the same number of epochs for the feature-level explanation. They all take around 10∼12 hours on a single NVIDIA RTX A6000 GPU, indicating the changes in trigger size and shape impose a minor influence on the runtime of SHINE.

**Ablation Study.** We add a comprehensive ablation study in the Pong game with the TrojDRL attack (we use the default trigger setup). Specifically, to verify the effectiveness of our trigger restoration. We first replace our trigger restoration method with FeatureRE (Wang et al., 2022) and apply our retraining method using the trigger restored by FeatureRE (Same as Exp-II.). Results in Tab. 9 show that SHINE is better than this baseline. We further verify the necessity of the step-level explanation. We directly apply the feature-level explanations using all the collected states and then retrain the agent with our proposed method. Tab. 9 (SHINE-NS vs. SHINE) shows that without the step-level explanation, we observe a performance drop in both the trigger fidelity and the agent's retraining performance. Note that we cannot remove the feature-level explanation as we need it to automatically pinpoint the trigger. To demonstrate the effectiveness of our retraining method, we replace it with

Table 7: Trigger detection fidelity of SHINE under different trigger patterns and trigger sizes.

| Dense Square Block | | | | Cross Sign | | | | Equal Sign | | | |
|---|---|---|---|---|---|---|---|---|---|---|---|
| $3 \times 3$ | $4 \times 4$ | $5 \times 5$ | $6 \times 6$ | $3 \times 3$ | $4 \times 4$ | $5 \times 5$ | $6 \times 6$ | $3 \times 3$ | $4 \times 4$ | $5 \times 5$ | $6 \times 6$ |
| 0.998 | 0.998 | 0.976 | 0.959 | 0.993 | 0.989 | 0.975 | 0.968 | 0.998 | 0.993 | 0.985 | 0.972 |

Table 8: Performance of the original poisoned agent and SHINE in the operating environment.

| Agent | Dense Square Block | | | | Cross Sign | | | | Equal Sign | | | |
|---|---|---|---|---|---|---|---|---|---|---|---|---|
| | $3 \times 3$ | $4 \times 4$ | $5 \times 5$ | $6 \times 6$ | $3 \times 3$ | $4 \times 4$ | $5 \times 5$ | $6 \times 6$ | $3 \times 3$ | $4 \times 4$ | $5 \times 5$ | $6 \times 6$ |
| Original | -0.010 | 0.021 | 0.035 | -0.082 | -0.031 | -0.024 | -0.021 | -0.039 | 0.019 | -0.024 | -0.091 | 0.030 |
| Retrained | 0.728 | 0.548 | 0.818 | 0.712 | 0.686 | 0.582 | 0.637 | 0.682 | 0.584 | 0.691 | 0.592 | 0.581 |

directly retraining the agent in the operating environment (Same as the direct retraining in Exp-II.). Tab. 9 (DR v.s SHINE) shows the effectiveness of our proposed retraining method.

Table 9: Ablation Study. "DR" stands for directly retraining. "SHINE -NS" means SHINE without the step-level explanation.

| Environment | Original | FeatureRE | DR | SHINE -NS | SHINE |
|---|---|---|---|---|---|
| Operating | -0.010±0.001 | 0.124±0.003 | 0.032±0.002 | 0.109±0.001 | 0.728±0.027 |
| Clean | 0.680±0.030 | 0.293±0.021 | 0.286±0.016 | -0.023±0.002 | 0.734±0.021 |

**SHINE in an Operating Environment without Clean States.** We consider a setup where the attacker poisons every state in the operating environment $P_\alpha = 1$. We used the Pong and breakout game for the experiment. The results are as follows. The retrained agent's reward in the operating/poisoned environment is: Pong: 0.805±0.032; Breakout: 30.26±1.102. The reward in the clean environment is: Pong: 0.705±0.023; Breakout: 20.380±1.026. Compared to the results in Table 2, we found that the retrained agent performs better in the operating environment. However, its performance in the clean environment drops from 0.734±0.021 to 0.705±0.023 in the pong game and from 25.350±1.609 to 20.380±1.026 in the breakout game. This is because, without clean states, we can only simulate clean states by masking out the trigger identified by our explanation method. Due to the inevitable approximation errors, the agent's performance in the clean environment drops slightly.

**Visualization.** Fig. 3 showcases the triggers and masks solved by SHINE for both backdoored and clean agents in the Pong game. In addition to the overall performance, we also take a closer look into the agent's action distribution in clean and poisoned states of the operating environment. Fig. 4 shows the action distribution of the backdoored agent before and after retraining in the Pong game. This result further explains the superior performance of SHINE, which enables almost identical distribution in clean and poisoned states.

## G    POTENTIAL SOCIAL IMPACT

Our method offers a practical tool to safeguard DRL agents deployed in many applications, enhancing their security and reliability. Additionally, it could potentially boost the policy-sharing market for large-scale RL models. By raising the bar for attackers to compromise DRL agents, our method can contribute to pushing the arms race between defenders and attackers in the field of DRL backdoor attacks. This can drive further advancements in defense techniques and make DRL systems more resilient to malicious attacks, benefiting the broader RL-related community.

