# OpenReview forum: "SHINE: Shielding Backdoors in Deep Reinforcement Learning"
_ICLR.cc/2024/Conference — Submitted to ICLR 2024_

### Official Review · Reviewer_2tnZ · 2023-10-31

**Soundness:** 2 fair
**Presentation:** 3 good
**Contribution:** 3 good
**Rating:** 6
**Confidence:** 4

**Summary:**

The paper proposes an explanation-based approach to defend against backdoor attacks in deep reinforcement learning. The main observation is that a backdoor attack aims to cause an RL agent to fail a game or get a very low total reward, which can be explained using the triggers present in the environment. To this end, the paper applies the self-explainable framework of Guo et al., 2021b, called EDGE, to identify a set of time steps where triggers are most likely to present and then extract a set of features from the state vector of these time steps and treat their average as triggers. The paper applies their approach to detect both perturbation-based backdoor attacks in single and multi-agent settings, where triggers are injected into states, and adversarial agent attacks, where a sequence of actions serves as triggers. It further proposes a backdoor shielding method to retrain the agent in the operating environment. The approach is evaluated using Atari games and the SMC environment for perturbation-based attacks and the MuJoCo environments for adversarial agent attacks.

**Strengths:**

Although various explanation-based defenses have been proposed for protecting deep learning, applying policy explanation techniques to protect RL agents from backdoor attacks seems new and promising. The approach does not require access to a clean environment or a set of clean trajectories.

The proposed approach applies to both perturbation-based attacks and adversarial agent attacks, outperforms several baselines in the former case, and obtains reasonable performance in the latter under some simple backdoor attacks against RL.

**Weaknesses:**

The paper is a direct application of the self-explainable framework for RL in Guo et al., 2021b, where the framework has already been applied to identify critical time steps associated with adversarial agent attacks. The technical contribution seems limited.

The proposed method relies on reward signals in the actual environment in its explanation component, which requires many failure trajectories to identify triggers. Hence, it is not applicable in security-critical domains.

I am not convinced that the approach can work in more challenging settings. To identify critical time steps associated with triggers using the EDGE framework, two assumptions are needed. First, there is a set of failure trajectories available. Second, the agent is supposed to win the game if the triggers are removed. However, a successful attack does not necessarily lead to a clear failure, especially in the case of stealthy attacks, where the goal could be reducing the agent's reward. Without domain knowledge of the expected performance of the agent in the target environment, it is hard to define what is considered a failure. Further, the evaluation considers a backdoor attack with a single fixed trigger placed at a fixed location, which is rather limited.

**Questions:**

How is (4) solved? Algorithm 1 only shows how to estimate the constraint but not how the shielding policy is optimized.

How are the set of trajectories used to train SHINE generated in the experiments? Are these all known to be failures?

The trigger detection stage of SHINE takes 12 hours, with an additional 5 hours used to retrain the policy. Is this comparable with other baselines? Since the approach requires many interactions with an infected environment, the agent will suffer from a significant loss until the trigger is mitigated. Hence, it seems unfair to only consider the reward after retraining when comparing the approach with other baselines.

Guo et al., 2021b show that significant improvement can already be achieved by partially blinding the victim agent's observation at the critical time steps in losing episodes. Thus, it would help to have an ablation study demonstrating the advantage of feature-level explanation and policy retraining.

In the experiment, it is assumed that there is a trigger in the environment with a probability of 0.1, 0.2, or 0.3. I wonder what would happen if triggers were present most of the time.

The evaluation uses a single fixed trigger pattern across RL training and testing. I wonder what would happen if the attacker varies the trigger used over rounds/episodes.

Table 2 shows that SHINE performs better than the original PPO in a clean environment. Why is this the case?

---

> ### Author Response · Authors · 2023-11-16
> **Response to Review 2tnZ**
>
> We thank the reviewer for the constructive comments. Please see below for our responses.
>
> **1. About the technical contributions.**
>
> Thank the reviewer for the comment. We agree with the reviewer that our method indeed uses the explanation method proposed by Guo et al., 2021b as it is the state-of-the-art step-level explanation method.  However, we would like to respectfully argue that this does not dilute our method’s technical contribution.  First, the overall defense framework that is composed of explanation-driven trigger restoration and retraining-based shielding is novel and is first proposed in this work. Second, rather than the step-level explanation, all the other technical components are newly proposed in this work. As such, we propose a new defense framework for backdoor defense with only one component leveraging an existing method. It is common that a newly proposed technique uses some state-of-the-art methods as part of its framework. We believe the proposed technique still demonstrates decent technical contributions.
>
> **2. About the proposed method requiring failure trajectories.**
>
> Thank the reviewer for the comment. As shown in the response to Review r1tp, we demonstrate that our method can work with only 20 failure trajectories. In addition, as stated in Section 5-Postmortem defense, we discuss that our goal is to quickly react to the backdoor attack once it is triggered. This is a common setup even for security-critical domains. For example, in cybersecurity, there is a directly called postmortem program analysis, which analyzes the reason and fixes the vulnerabilities after it is triggered. As is also discussed in Section 3.1, in practice, it is also difficult to guarantee the environment is clean. For example, in self-driving cars, it is extremely hard to guarantee that the attacker cannot access the environment where the car is deployed. As such, it is possible that the attacker will trigger a backdoor and cause some damage. Our goal is to react quickly and make sure similar damage will not happen in the future.
>
> **3. About the application of our method in other settings.**
>
> Thank the reviewer for the comment.  The reviewers pointed out two setups where our defense may not be applicable. First, the reviewer mentions that a successful attack does not necessarily lead to a clear failure. We would like to respectfully mention that as a defense work, we follow the setups in existing attacks. We believe this is the case for most defenses (not only backdoor defenses in DRL). As stated in the attack papers (TrojDRL and backdoorl), for games with a discrete final reward (win or lose), the attack goal is to make the victim agent lose the game. For games with continuous rewards, the attack goal is to reduce the victim agent’s total reward. As stated in Eqn. (1), the step-level explanation considers both cases. Actually, as shown in Table 2, the attacks on Atari games reduce the victim agent’s reward (this is no clear win or lose). Our method can still maintain its effectiveness in such cases.
>
> Second, the reviewer mentions that the attack can change the trigger’s location. Again, we would like to respectfully point out that this is the setup considered in existing attacks. In Section-5 adaptive attacks, we indeed try to change the trigger location as it can be an adaptive attack against our defense. However, the attack is not successful. This result demonstrates that our defense is effective for state-of-the-art attacks, which is enough to demonstrate its efficacy. As the attack and defense, in general, is a cat-and-mouse game. Our defense and this fail adaptive attack motivate the design for stronger attacks. We believe this can be an interesting future work. However, it is not the focus of this paper.
>
> **4. How to solve the restraining objective function and how to collect trajectories for the step-level explanation.**
>
> Thank the reviewer for the comment. First, we solve Eqn. (4) via gradient descent. We follow the procedure as the PPO algorithm, with a different set of constraints. We will clarify this in the next version. Second, we collect the trajectories of the given policy by running it in the operating environment where the trigger will show up. We will collect the state, action, and reward for each trajectory for applying the explanation method. We will clarify this in the next version.

---

> > ### Author Response · Authors · 2023-11-16
> > **Response to Review 2tnZ**
> >
> > **5. About the computational cost, loss during the mitigation, and metric.**
> >
> > Thank the reviewer for the comment. Regarding the computational cost, all of the three baseline defenses (NC, FeatureRE, and Direct retraining) require retraining. This part of the cost is the same for all methods.
> > Regarding trigger restoration, Direct retraining does not conduct this step and thus does not have a cost. NC and FeatureRE take 12 and 15 hours for this step. Their restored trigger is of very low fidelity, and they can only be applied to environments with a discrete action space.  We will clarify this in the next version.
> >
> > Given that all three baseline methods also require retraining in the operating environment to remove the backdoor, they also introduce losses at this stage. This is the same for all methods. If a simulator is available, this step can be done in the simulator, which can reduce the loss.
> >
> > Given that all three baseline methods and our method require retraining. In addition, the goal of the defense is to improve the agent’s performance when the trigger is shown up in the environment while maintaining the agent’s performance in the clean environment. We believe the reward after retraining in the clean and operating environment is the proper metric for evaluating the effectiveness of the defenses.
> >
> > **6. About the ablation study.**
> >
> > Thank the reviewer for the comment. In Appendix-F-Other experiments-Ablation study, we conduct an ablation study to demonstrate the effectiveness of our feature-level explanation and the retraining technique.
> >
> > **7. About the trigger show-up probability.**
> >
> > Thank the reviewer for the comment. In Appendix-F-Other experiments-SHINE in an Operating Environment without Clean States, we demonstrate that our method is effective if the trigger show-up probability equals 1.
> >
> > **8. Why do shielded agents perform better than the original agent in a clean environment.**
> >
> > Thank the reviewer for pointing this out. We suspect this is because the retraining process improves the generalizability of the agent as the operating environment can be taken as a variation of the original agent. Similar results are also observed in the adversarial example attacks against deep neural networks. That is, adversarial retraining can sometimes improve the performance of the classifier on clean inputs [1]. In addition, our retraining constraint in Eqn. (4) constrain the value function of the retrained policy to be similar rather than to be smaller than the value function of the original policy in the clean environment. It is possible that the retrained policy performs even better than the original policy in the clean environment.
> >
> > [1] Explaining and harnessing adversarial examples

---

> > ### Comment · Reviewer_2tnZ · 2023-11-19
> >
> > Thanks for the clarifications. While the Atari results make sense, it's still unclear when and how the solution should be applied in practice for environments without a clear definition of failures. The experiments assume that there are always triggers in the chosen trajectories. What if the assumption is wrong? In particular, the agent may experience failures or low returns with a certain probability even in a clean environment without triggers. What would happen if the algorithm is applied to this case?

---

> > > ### Comment · Reviewer_2tnZ · 2023-11-19
> > >
> > > It is also unclear why the explanation-based solution can work when no clean states exist. How many failure trajectories and winning trajectories are there in the experiment? Do the winning trajectories have any clean states?

---

> > > > ### Author Response · Authors · 2023-11-20
> > > > **Response to Reviewer 2tnZ**
> > > >
> > > > Thanks the reviewer for the additional question regarding why the explanation-based solution can work when no clean states exist. In our experiment, If there is no clean state, the collected trajectories will all be failure trajectories (we collect 100 of them).
> > > > Here, our step-level explanation can still be used to fit the collected trajectories and identify some important states. The step-level explanation may not be entirely faithful. Given that all the states contain the trigger, we can still apply our feature-level explanation method to the identified states and thus pinpoint the trigger. Actually, the trigger restoration becomes easier because we rely less on the step-level explanation method to pinpoint states with the trigger, given all states have the trigger presented.
> > > >
> > > > Again, we thank the reviewer very much for the questions. We will clarify them in the next version. We are happy to answer follow-up questions. Meanwhile, we respectfully and kindly ask the reviewer if they can reconsider their score if our response helps address the reviewer's concerns.

---

> > > > > ### Comment · Reviewer_2tnZ · 2023-11-23
> > > > >
> > > > > Thanks for the clarifications. It seems that the approach heavily relies on the fact that the attacker uses a fixed trigger. Otherwise, it does seem possible to identify the trigger using the failure trajectories only without any clean states. Thus, I'm still not convinced that the approach will work against more advanced attacks. That being said, I do see the potential of the explanation-based approach and appreciate the authors' effort in adding experiment results and providing detailed justifications. I'll increase my score.

---

> > > ### Author Response · Authors · 2023-11-20
> > > **Response to Reviewer 2tnZ**
> > >
> > > We thank the reviewer 2tnZ for the kind response and the additional question.
> > >
> > > First of all, we would like to respectfully and kindly state that as a defense work, we follow the threat model/attack assumptions of existing backdoor attacks. All existing backdoor attacks define a successful attack as significantly reducing the victim agent's reward when the trigger is presented in the environment, which is a failure. Our defense leverages this definition to explain the failure cases and identify the trigger. If there is no clear definition of failures, first, it is kind of difficult to define a successful attack. Even in such cases, our step-level explanation can still work. As shown in Eqn. (1), the explanation can handle the case where the reward design is either continuous or discrete. We can assume a case where the attack is not that successful, and it causes the total reward to reduce but not dramatically. This should follow the setup that there is no clear definition of failures. Here, the step-level explanation can still applied to fit the collected trajectories and identify the trigger. If the reward is continuous, we can capture the reward difference naturally. If the reward is discrete, by setting a small range for each class in the categorical distribution (Eqn. (1)), our explanation method can capture the relatively non-significant reward difference. Again, we would like to kindly and respectfully mention that although SHINE can be potentially applied to such cases, our main focus is still the threat model of existing attacks (i.e., trigger significantly reduces the victim agent's reward). We will further clarify this in the next version.
> > >
> > > We agree with the reviewer that there could be some trajectories where the failure is not caused by the trigger. First, SHINE works in the operating environment where the trigger will show up. If the trigger is never presented, it belongs to a different threat model from us. As such, the failure trajectories will have trajectories caused by the trigger (with the trigger) and trajectories caused by other factors (without the trigger). Our feature-level explanation will identify a dense patch or a subset of features in the state space as the trigger. We will add the elastic net regularization when solving the trigger. This will only identify the trigger. We agree that so far, our defense cannot deal with attacks using a watermark as the trigger (as discussed in Section 5). But for the dense triggers used in existing attacks and their variations, we can handle them.

---

### Official Review · Reviewer_F9HD · 2023-10-31

**Soundness:** 2 fair
**Presentation:** 3 good
**Contribution:** 3 good
**Rating:** 5
**Confidence:** 5

**Summary:**

In this In this paper, the authors study the problem of defense against backdoor attack in deep reinforcement learning. They present a practical algorithm called SHINE that first identifies backdoored features in attacked states and then sanitizes them in the hope of rendering the backdoor behavior ineffective on the agent. They claim that their methods theoretically improves the backdoored agent’s performance under attack while still retaining its performance in clean environment. They also test their method on three benchmark deep RL environments and show its effectiveness in eliminating backdoor attacks.

**Strengths:**

1. This paper studies an important problem of safeguarding against adversarial attacks in deep RL.
2. Their algorithm is well motivated and the authors claim it to work against tested benchmark environment.

**Weaknesses:**

1. No formal definition of attack and defense is provided. It is not even clear when does attacker attack, does it use any attack policy? The notations used in section 3.3 is not sufficiently clear.

2. In adversarial agent attack, how are actions embedded in state as you claim in the beginning of page 5? Actions usually come from a different space of discrete state than states.

3. This is perhas a good empirical paper, but it is not sound theoretically. I would formulate the problem properly and tune down the emphasis to adversarial agent attack part and theoretical claims to make a good case for it. It is necessary to define the assumptions under which the theoretical claims are guaranteed to work.

4. I believe for sanitization using masking approach to work, it needs certain assumption on kind of triggers that adversary can be put in environment and it is not clear from the paper why your approach would work in a general case. See question 4 for an example.

**Questions:**

1. Learning a feature explanation mask for each pixel in a state does not look very scalable especially in environments like Atari games where state space could get really large. How do you address this problem?

2. What is local linear constraint in line above Feature-level explaination paragraph? How does it help?

3. Without any assumption, it may happen that the adversary may not attack when the algorithm is run. Does the algorithm still work?

4. As a case study, let say the adversary designs a patch trigger to be put on top right corner of the state image(say in ping pong game). Normally, the pixel value is uniformly 100 and the adversary has trained the policy so that when pixel value is 255 or 0, it takes backdoor action. While attacking the adversary only inject 255 pixel patch. Your method of simply deleting the feature would lead to a zero patch which is adversarial as well. How would you fix this?

---

> ### Author Response · Authors · 2023-11-16
> **Response to Review F9HD**
>
> We thank the reviewer for the constructive comments. Please see below for our responses.
>
> **1. About the definition of the attacks and defenses.**
>
> Thank the reviewer for the comment. As stated in the introduction, the existing attacks against DRL policy train a victim policy $\pi$ for an agent in a given environment. The agent performs normally in the clean environment, i.e., will finish its corresponding task with a high success rate and achieve a high reward. The victim policy will take suboptimal actions when the trigger shows in the environment state. This will lower the victim agent's reward.
>
> Specifically, perturbation-based attacks add a small patch as the trigger to the environment state and the suboptimal actions of the victim policy can be a random action (Non-targeted attacks) or a specific action (targeted attacks). As stated in the evaluation section, we follow the original attack papers and add the trigger patch to each state with a certain probability.
>
> Adversarial-agent attacks target two-player competitive games with an adversarial agent and a victim agent. The trigger is a sequence of continuous actions of the adversarial agent. Once the adversarial agent takes the trigger actions, the victim agent will perform suboptimally. As stated in the evaluation, we follow the original attack paper and let the adversarial agents take the trigger action at the start of a game.
> We are sorry for the confusion. We will summarize these attack descriptions in one paragraph in the next version.
>
> Regarding our defense, we are given a shielding agent’s policy $\pi$ and identify a trigger $\tau$ at the trigger restoration step (Section 3.2). Then, we use the restored trigger to retrain the policy and obtain a shielded policy $\hat{\pi}$.
>
> **2. About Adversarial agent attacks.**
>
> Thank the reviewer for the comment. We follow the setups and arguments in the backdoorl paper. For MuJoCo environments, the action space of both agents is continuous rather than discrete. Second, in the state representation of one agent, there are certain dimensions corresponding to the other agent’s action. As such, the actions of one agent are embedded in the state representation of the other agent. We consider MuJoCo environments because this is the only environment in which backdoorl apply their attacks.
>
> **3. Questions about the explanation methods.**
>
> Thank the reviewer for the questions. First, our proposed feature-level explanation learns a mask matrix for all pixels in the state representation, where all the individual masks are optimized together. As such, our method can handle environments with high-dimensional state representations, such as Atari games (as shown in our evaluation).
>
> Regarding the local linear constraint in the step-level explanation (EDGE), it is inherent in the AlvarezMelis & Jaakkola (2018) paper. It helps make the explanation method perform locally linear around the given samples, which improves the explainability of the approximation model. It is equivalent to conducting a piece-wise linear approximation of a complicated non-linear function, where the piece-wise linear function is explainable. We will clarify this in the next version.
>
> **4. About the assumptions of SHINE.**
>
> Thank the reviewer for the comment. As stated in Section 3.1 threat model, SHINE does require the attack launch the attack. We do not consider the setup where the defense is only given a clean environment without the attack being triggered. We also follow the existing attacks and consider the case where the attacker adds one backdoor with one trigger in the victim policy. This is an implicit assumption inherited from the attack setups. Similarly, given our mask strategy, we do not consider the cases where the attacker uses zero pixels as the trigger for perturbation-based attacks (which are also not used in the existing attacks). We will further clarify these assumptions in the next version.

---

> > ### Author Response · Authors · 2023-11-16
> > **Response to Review F9HD**
> >
> > **5. About the question of our masking.**
> >
> > Thank the reviewer for the comment. We agree with the reviewer that if the attackers use zero pixels as the trigger, our masking strategy may not work as it is equivalent to adding the trigger to the environment. A solution to this is to replace the detected trigger with mean pixels across all collected states. To avoid the case where the trigger consists of the mean pixels, we can further apply a Gaussian blur on the mean pixels and then replace it with the detected trigger. We apply this solution to two attack setups on the Pong game (3$\times$3) where the first attack uses the zero pixels as the trigger (denoted as attack-1) and the second attack uses the mean pixels across $1000$ states as the trigger (denoted as attack-2). The performance of our shielded agents is attack-1: 0.725 (in the operation environment)/0.731 (in the clean environment); attack-2: 0.727 (in the operation environment)/0.732 (in the clean environment). This result is comparable with our original evaluation results in Section 4, which demonstrates the effectiveness of this solution. We thank the reviewer for pointing this out and will add the solution to the next version. Note that our explanation will pinpoint the actual trigger. If we find the trigger is all zero or mean pixels, we can use this solution rather than masking. For other cases, we can still apply masking.

---

> > > ### Author Response · Authors · 2023-11-20
> > > **Follow up with the reviewer F9HD**
> > >
> > > We thank the reviewer very much for the questions and comments. We would like to kindly follow up with the reviewer. We are happy to answer follow-up questions. Meanwhile, we respectfully and kindly ask the reviewer if they can reconsider their score if our response helps address their concerns.

---

> > ### Comment · Reviewer_F9HD · 2023-11-21
> >
> > I have read the rebuttal and am not very satisfied by the response. Specifically, the paper still lacks a formal attack and defense setup and it is not clear under what assumptions on attack will their method work. So, I would keep my score unchanged.

---

> > > ### Author Response · Authors · 2023-11-21
> > > **Response to the reviewer**
> > >
> > > Thanks the reviewer F9HD for the additional comment. We have responded in points 1&4, a clear setup of our attack and defense, together with our assumption. In point 5, we responded to the corner cases mentioned by the reviewer. It would be super helpful if the reviewer could point out which part of the setup and assumption is still not clear. Currently, we have made it clear that our defense is effective against both perturbation-based attacks and adversarial agent attacks under single- and multi-agent environments. We follow the attack setups of all three main-stream backdoor attacks against DRL. That is, the attacker adds a trigger (either a patch or a sequence of actions) to the environment will significantly reduce the total reward of the victim agent collected in one episode. The patch can be all zero (as pointed out in response point 5). The additional assumption our defense requires is the trigger needs to be presented. We really appreciate it if the reviewer could point out more specifically which part is still clear. Thank you very much!

---

> > > > ### Comment · Reviewer_F9HD · 2023-11-23
> > > >
> > > > I meant a formal mathematical definition of attack and defense using the MDP framework of RL. The notion of value and how attack and defense affects this value that the agent cares about is also not clearly defined. Please take a look at the setup of [1] for a formal definition of the attack policy $\nu$ and value the agent gets under attack.
> > > >
> > > > [1]. Robust Deep Reinforcement Learning against
> > > > Adversarial Perturbations on State Observations https://arxiv.org/pdf/2003.08938.pdf

---

### Official Review · Reviewer_5eAS · 2023-10-31

**Soundness:** 3 good
**Presentation:** 3 good
**Contribution:** 3 good
**Rating:** 6
**Confidence:** 2

**Summary:**

This paper addresses the vulnerability of deep reinforcement learning (DRL) policies to backdoor attacks. It introduces SHINE, a method that combines policy explanation techniques and policy retraining to mitigate these attacks effectively. The proposed method is theoretically proven to enhance the performance of backdoored agents in poisoned environments while maintaining performance in clean environments. Experiments demonstrate its superiority over existing defenses in countering DRL backdoor attacks.

**Strengths:**

- This paper studies the problem of backdoor threats in reinforcement learning. The authors leveraged several techniques to identify the backdoor triggers and designs a policy retraining algorithm to eliminate the impact of the triggers on backdoored agents.
- Theoretical guarantees in terms of the performance are provided.
- Empirical evaluations are promising.

**Weaknesses:**

- It would be helpful to test the proposed methods against a broader set of backdoor attacks.
- Could you also compare your method with other types of defenses, e.g. non-trigger-inversion?

**Questions:**

-

---

> ### Author Response · Authors · 2023-11-16
> **Response to Review 5eAS**
>
> We thank the reviewer for the constructive comments. Please see below for our responses.
>
> **1. Compare SHINE against other types of defenses.**
>
> Thank the reviewer for the comment. We follow the reviewer’s comment and add one more baseline defense: STRIP [1]. This defense assumes the inputs containing the trigger, and it tries to pinpoint the trigger. It does not conduct trigger inversion (like FeatureRE or NE). We use this method to detect the trigger for perturbation-based attacks (given that it can only be applied to games with discrete action spaces). After detecting the trigger, we apply our retraining method to remove the backdoor.  We selected the pong environment for attacks against single-agent RL and the QMIX setup for attacks against multi-agent RL. The results in the following table show that SHINE is better than this method. In addition, SHINE can apply to adversarial agent attacks, while STRIP cannot.
>
> |           | Method | Pong  | QMIX  |
> |-----------|--------|-------|-------|
> | Fidelity  | SHINE  | 0.998 | 0.936 |
> | Fidelity  | STRIP  | 0.853 | 0.811 |
> | Operating | SHINE  | 0.728 | 99.1  |
> | Operating | STRIP  | 0.624 | 85.3  |
> | Clean     | SHINE  | 0.734 | 99.2  |
> | Clean     | STRIP  | 0.612 | 87.5  |
>
> [1] STRIP: A Defence Against Trojan Attacks on Deep Neural Networks
>
>
> **2. Evaluate SHINE against broader attacks.**
>
> Thank the reviewer for the comment. In our evaluation, we have evaluated our method against all three existing mainstreaming backdoor attacks in RL. These attacks cover both single-agent and multi-agent RL. They also consider different types of triggers: state perturbations and adversarial agents. We also consider different attack variations and adaptive attacks. With this large set of attacks, we humbly believe that our evaluation has already covered most attacks against RL and is able to demonstrate the effectiveness of our defense. If the reviewer believes we need to evaluate SHINE against other specific attacks, we are happy to add those additional evaluations.

---

> > ### Author Response · Authors · 2023-11-20
> > **Follow up with reviewer 5eAS**
> >
> > We thank the reviewer very much for the questions and comments. We would like to kindly follow up with the reviewer. We are happy to answer follow-up questions. Meanwhile, we respectfully and kindly ask the reviewer if they can reconsider their score if our response helps address their concerns.

---

> > ### Comment · Reviewer_5eAS · 2023-11-22
> > **Thanks for the rebuttal**
> >
> > I would like to thank the authors for the additional experiments. The results look promising and hence I will increase my score.

---

> > > ### Author Response · Authors · 2023-11-22
> > > **Thank the reviewer 5eAS for the positive feedback**
> > >
> > > Thank the reviewer 5eAS for the positive feedback. We will integrate the new experiments in the next version.

---

### Official Review · Reviewer_r1tp · 2023-11-03

**Soundness:** 3 good
**Presentation:** 2 fair
**Contribution:** 3 good
**Rating:** 6
**Confidence:** 3

**Summary:**

This paper proposed an algorithm dubbed 'SHINE', which is a testing-phase shielding method against backdoor attacks in the context of Deep Reinforcement Learning (DRL). SHINE first captures the critical states related to the backdoor trigger based on DRL explanation methods, then identifies a subset of features using the proposed feature-level interpretor. The agent policy is re-trained after trigger identification. Empirical evaluation verifies the efficacy of SHINE in shielding backdoored agents against different backdoor attacks.

**Strengths:**

\+ This paper tackles a challenging and important topic of shielding DRL agents.

\+ Wide applicability: the proposed approach can defense against both perturbation-based attacks and the adversarial agent attacks, for either single-agent or multi-agent RL.

\+ The idea of pinpointing crucial states -> identifying triggered features -> retraining policy is well motivated and technically sound.

\+ Experimental design is comprehensive regarding trigger identification, shielding effectiveness, and performance impacts on clean environments. Sensitivity analysis is well designed.

**Weaknesses:**

\- This method hinges on the access to corrupted trajectories and environments for identifying the trigger. Sensitivity analysis is necessary to investigate whether the number or distribution of of those trajectories affect the efficacy of the proposed method.

\- Related work: The related work section could use some more efforts in elaborating how adversarial based attack works, as well as more prior work on the DRL explanation.

\- Need more elaboration on how step and feature-level explanation would be applied if attack is adversarial based, and how multi-agent RL can benefit from this approach.

\- Need more explanation on the two constraints in Eq (4), which is the key of the re-training process. In general the writing of the Sec 3.3 is too abstract to derive to the final objective.

**Questions:**

-  Does the number of pre-trained trajectories matter to the shielding capability?
- How this method applies to Multi-agent DRL scenarios?
- It is not clear to me why $M_\pi(\hat{\pi}) \geq M_\pi(\pi)=\hat{\eta}(\pi)$.

---

> ### Author Response · Authors · 2023-11-16
> **Response to Review r1tp**
>
> We thank the reviewer for the constructive comments. Please see below for our responses.
>
> **1. Sensitivity analysis regarding the impact of the number of trajectories used for explanation.**
>
> Thank the reviewer for the comment. Following the reviewer’s comment, we conducted an additional experiment to evaluate the impact of the number of trajectories on our method. We used the Pong game for this experiment. We vary the number of trajectories as 200/500/1000 and report the trigger fidelity and final retraining performance. The results in Table 1 show that our method is not that sensitive to this factor. It also shows that SHINE does not require extensive failed cases to apply shielding. In addition, as stated in Section 5-Postmortem defense, we discuss that our goal is to quickly react to the backdoor attack once it is triggered.
> | # of trajectories             | Trigger fidelity | Operating | Clean |
> |-------------------------------|------------------|-----------|-------|
> | Failure: 100 and Succeed: 900 | 0.998            | 0.728     | 0.734 |
> | Failure: 50 and Succeed: 450  | 0.991            | 0.721     | 0.729 |
> | Failure: 20 and Succeed: 180  | 0.978            | 0.698     | 0.709 |
>
> **2. Questions about the related work.**
>
> Thank the reviewer for the comment. The review first asked for more description of “ how adversarial based attack works.” We would like to kindly confirm with the reviewer if the reviewer referred to the “adversarial-agent attack” or other attacks. The adversarial agent attack is designed for two-agent competitive games, where one agent is the adversarial agent that takes the trigger action, and the other is the victim agent that contains the backdoor. At a high level, this attack designs the trigger as a sequence of continuous actions of the adversarial agent. Then, it constructs two policies for the victim agent. One policy is a normal policy, and the other is a policy that reacts to the trigger actions and fails the task. Finally, it fuses these two policies into one policy network via behavior cloning and uses it as the victim agent’s policy. We will add this to the next version (Please let us know if the reviewer referred to some other attacks by saying “adversarial based attack”). The reviewer also suggests providing more literature about the DRL explanation. We had some related descriptions in the Section 3. We will follow the reviewer’s suggestion and provide more literature review in Section 2.
>
> **3. Questions about the generalizability.**
>
> Thank the reviewer for the comment. Again, we would kindly ask for the reviewer to explain a little bit more concretely what the “attack is adversarial based” refers to. If it refers to adversarial agent-based attacks, we evaluated such attacks in the evaluation. Since we target trojan attacks, which are launched by the attacker, they are naturally adversarial attacks. In our evaluation, we also evaluate our defense against the state-of-the-art backdoor attacks in multi-agent RL and demonstrate that SHINE is still effective against this attack. In Appendix A.4, we explained how we extend SHINE to this attack. At the high level, we shield each backdoored agent in turn. We just need to change the input to explanation methods (local or global observation) based on the agents’ training methods.
>
> **4. Questions about the constraints in Eqn (4).**
>
> Thank the reviewer for the comment. The first constraint ($\hat{K}$) constrains the agent’s performance change in the poisoned (operating) environment. This is the constraint used on the PPO algorithm when updating the policy. Using it together with the optimization objective function in Eqn. (4) enforces the agent to achieve a better performance in the poisoned environment (removing the backdoor). The second constraint ($K$) comes from Theorem 2. It constrains the agent’s performance change in the clean environment. This constraint helps maintain the agent’s performance in the clean environment (maintaining the agent’s utility, i.e., clean performance). $M_{\pi}(\hat{\pi}) \leq M_{\pi}(\pi)$ because we maximize $M_{\pi}(\hat{\pi})$, so the $M_{\pi}$ of the updated policy $\hat{\pi}$ is at least the same as the $M_{\pi}$ of the previous policy $M_{\pi}(\pi)$. $M_{\pi}(\pi) = \hat{\eta}(\pi)$ because the KL term equals zero, and the advantage function is also zero when $\hat{\pi} = \pi$. We will follow the reviewer’s comment and make this part more clear in the next version.

---

> > ### Author Response · Authors · 2023-11-20
> > **Follow up with reviewer r1tp**
> >
> > We thank the reviewer very much for the questions and comments. We would like to kindly follow up with the reviewer. We are happy to answer follow-up questions. Meanwhile, we respectfully and kindly ask the reviewer if they can reconsider their score if our response helps address their concerns.

---

### Meta-Review · Area_Chair_1S8K · 2023-12-09

**Metareview:**

The paper introduces SHINE, an innovative approach for shielding deep reinforcement learning (DRL) agents from backdoor attacks. It leverages policy explanation techniques and a policy retraining algorithm to identify and mitigate backdoor triggers, showing improvement in poisoned environments while maintaining performance in clean ones.

Extensive experiments validate its effectiveness against various DRL backdoor attacks. However, discussions with reviewers highlighted some issues, such as its reliance on specific types of corrupted data, the need for more comprehensive exploration of related works, and a lack of clarity in certain aspects. Although the authors responded proactively to these concerns, providing additional experiments and clarifications, some reviewers remained unconvinced about the scalability and generalizability of the method, particularly in more advanced or varied attack scenarios.

For future improvements, it's recommended that the authors focus on enhancing the clarity and formal definitions of attacks and defenses, expand the scope of related work, and explicitly state and justify the assumptions underpinning their method to improve its relevance and applicability to diverse real-world scenarios.

**Justification For Why Not Higher Score:**

Reviewers’ remaining concerns about the clarity of formal attack and defense setups within the DRL framework, and the scalability of the method in handling more complex attack patterns, suggest the need for further refinement in these areas.

**Justification For Why Not Lower Score:**

The paper proposes a novel approach in addressing a critical area of DRL security, with road applicability across a range of attack types, and robust experimental validation. The authors' theoretical grounding and responsiveness to reviewers' queries add to its merits.

---

### Decision · Program_Chairs · 2024-01-16

Reject